# Comparative and Correlation Analysis of Young and Mature Kaffir Lime (*Citrus hystrix* DC) Leaf Characteristics

Rahmat Budiarto [1,*], Roedhy Poerwanto [2], Edi Santosa [2], Darda Efendi [2] and Andria Agusta [3]

1 Department of Agronomy, Faculty of Agriculture, Universitas Padjadjaran, Sumedang 45363, West Java, Indonesia
2 Department of Agronomy and Horticulture, Faculty of Agriculture, Institut Pertanian Bogor, Bogor 16680, West Java, Indonesia
3 Research Center for Pharmaceutical Ingredient and Traditional Medicine, National Research and Innovation Agency, Cibinong 16911, West Java, Indonesia
* Correspondence: rahmat.budiarto@unpad.ac.id

**Abstract:** Kaffir lime is leaf-oriented minor citrus that required extra attention to study. This study aimed to (i) comparatively analyze the young and mature leaf morpho-ecophysiological characters; and (ii) perform a correlation analysis for revealing the relationship among the physiological characters. Plants were ten one-year-old kaffir lime trees cultured under full sun condition. Leaf size was measured by using a specific allometric model. The Li-6400XT portable photosynthesis system was used to observe the leaf ecophysiological characters. The statistical analysis revealed significant differences in leaf size and physiology as the effect of leaf age. A significant size enlargement in mature leaves was noticed, especially in terms of leaf length, area, and weight, of about 77%, 177%, and 196%, respectively. Young leaves experienced a significant improvement in photosynthetic rate and actual water use efficiency for about 39% and 53%, respectively. Additionally, a strong, significant, and positive correlation between leaf chlorophyll, carotenoid content, and photosynthetic rate was found in the present study. Further studies using a multi-omics approach may enrich the science between kaffir lime leaf maturation as the basis of agricultural modification practice.

**Keywords:** *Citrus hystrix* DC; leaf age; Li-6400XT; photosynthetic rate; water use efficiency

## 1. Introduction

Citrus, an ancient crop [1], has become one of the most important horticulture commodities worldwide, with a quadruple production gain during the 1960s to 2012 [2]. The latest genomic study revealed the Southeastern Himalayan valley as citrus original site, before eventually spreading around the world by numerous human activities [3]. Southeast Asian countries (Vietnam, Thailand, and Indonesia) are citrus (lime) exporters to the world market [4] and has become the center of diversity for kaffir lime [5,6].

Unlike fruit-oriented citrus in general, the kaffir lime is more popular for its leaves [7–9] in Asian cooking recipes [9–11] and for essential oils [12–14]. In the future, the uses of kaffir lime leaf as a cooking spice and essential oil would increase, along with the increase in population growth. Numerous studies have proved various pharmacological properties of citrus essential oil, such as stress relief and sleep relaxation [15], larvacidal [16], antifeedant [17], antibacterial [18], antifungal [19], antiparasitic [20], antimicrobial [21], anticancer [22], anti-inflammatory [23], and antioxidant [24]. Due to its economic importance, its production should meet the demand. Strategies to gain agri-production can be land expansion and or intensifying of agri-inputs, i.e., seedling, fertilizer, irrigation, and pest control agents [25].

To gain kaffir lime production means to produce more leaves. Leaves serve as the main source organ, where the photosynthesis process takes place [26,27], whose growth patterns can go simultaneously or alternately with the root part [28]. Due to the effect of leaf growth, an early study classified its functionality to be young and mature leaves [29].

Different leaf ages may display different leaf morpho-physiological characters. Leaf age is an important variable to study due to its association with energy partition [30] and pest control issues [31–33]. Previous studies [34,35] have been described the effect of leaf age on a plant in general. However, there is still limited research specific to the leaf morpho-physiological characteristics of a leaf-oriented citrus variety such kaffir lime. Therefore, the present study aimed to (i) comparatively analyze the leaf morpho-ecophysiological character as the leaf is getting mature; and (ii) perform a correlation analysis for revealing the relationship among the physiological characters.

## 2. Materials and Methods

### 2.1. Study Site and Plant Material

The present study was carried out in the Pasir Kuda experimental garden, IPB University, Bogor, Indonesia (6°36′36″ S, 106°46′47″ E, 239 m above sea level), from March 2018 to March 2019. Ten one-year-old kaffir lime trees cultured on latosol soil and under tropical open-field conditions were used as the plant materials in the present study. The plants, originated from grafted seedling into Rangpur lime rootstock, were raised by fertilizer application (Dose per plant: 20 g N, 15 g $P_2O_5$ and 10 g $K_2O$), hand weeding, and insecticide application (if necessary).

### 2.2. Research Procedure

A single young and mature leaf was selected in each tree; thus, there were 10 young leaves and 10 mature leaves involved in the present study. Due to the completely randomized design used by the present experiment, the selection of the targeted leaf was random in numerous leaves in the kaffir lime canopy, as long as the targeted leaf is pest and disease damage free and display a normal bifoliate form. In general, the one-year-old kaffir lime trees were ± 1 m in height and had 120–150 leaves.

### 2.3. Measured Variables

Variables of leaf size, such as leaf length, leaf area, and leaf weight, were measured by using an allometric approach specific to Indonesian kaffir lime [7]. Leaf ecophysiological characteristics were measured by using a Li-6400XT portable photosynthesis system (Licor Inc., Lincoln, NE, USA) in the sunny day at 9:30 a.m., 7 March 2019. A Li-6400XT was able to measure the rate of transpiration (mmol $H_2O$ $m^{-2}$ $s^{-1}$), the rate of photosynthesis (µmol $CO_2$ $m^{-2}$ $s^{-1}$), stomatal conductance (mol $H_2O$ $m^{-2}$ $s^{-1}$), incoming radiation (W $m^{-2}$), and leaf temperature (°C). In every single measured leaf, there would be three measurements (triplo) made automatically by the Li-6400XT, defined as three observational replications. In addition, the actual water-use efficiency (WUE) of the kaffir lime leaf was calculated by dividing the obtained photosynthetic rate to the transpiration rate and then expressed in µmol $CO_2$ mmol $H_2O^{-1}$. The actual light-use efficiency (LUE) was calculated by dividing the leaf fresh weight to perceived sunlight and then expressed in µg $lux^{-1}$. The pigment content was measured by using a spectrophotometer following a previous study [36].

### 2.4. Data Analysis

Collected data were then subjected to the least significant difference (LSD) test at α 5% in Statistical Analysis Software (SAS) version 9.4 (SAS Institute Inc, Cary, NC, USA). Correlation analysis was performed between the photosynthetic rate and leaf pigment content (chlorophyll-a, chlorophyll-b, chlorophyll total, carotenoid and anthocyanin) by using Statistical Tool for Agricultural Research (STAR) version 2.0.1. (IRRI, Los Banos, Philippines).

## 3. Results and Discussion

### 3.1. Comparative Analysis of Leaf Morphological Character

Leaf size is an important morphological character that has been frequently observed in numerous papers. The area of leaf is aimed to evaluate the accumulation of plant biomass [37] and also the leaf growth [38–40]. In addition, leaf size was also the basis for

considering the suitability of ornamental citrus potted plant that is widely developed in green cities [41,42]. The leaf weight was also used for similar purposes, with an addition to know the leaf harvesting index [43]. Leaf length was previously reported to be used as input in a non-destructive allometric model to estimate leaf area and leaf weight on kaffir lime [7].

Statistical analysis in the present study showed that young leaves experienced a significant size enlargement as the leaf is getting older. Young leaf length was 6.20 cm, while the mature one had already 10.99 cm. The mature leaf area was 29.18 cm$^2$, while the young one was only 10.53 cm$^2$. Individual leaf weight was recorded at 0.30 g and 0.88 g during its young and mature stage, respectively. The weight, area, and length of the young leaves were significantly increased up to 196%, 177%, and 77%, respectively, when it turned into the mature phase (Figure 1a–c). As the leaf maturation process continued the young leaf was getting broader to the maximum size [44,45]. The variation in leaf size might lead to different physiological responses, since leaf area is associated with the light absorption site on plants [46,47].

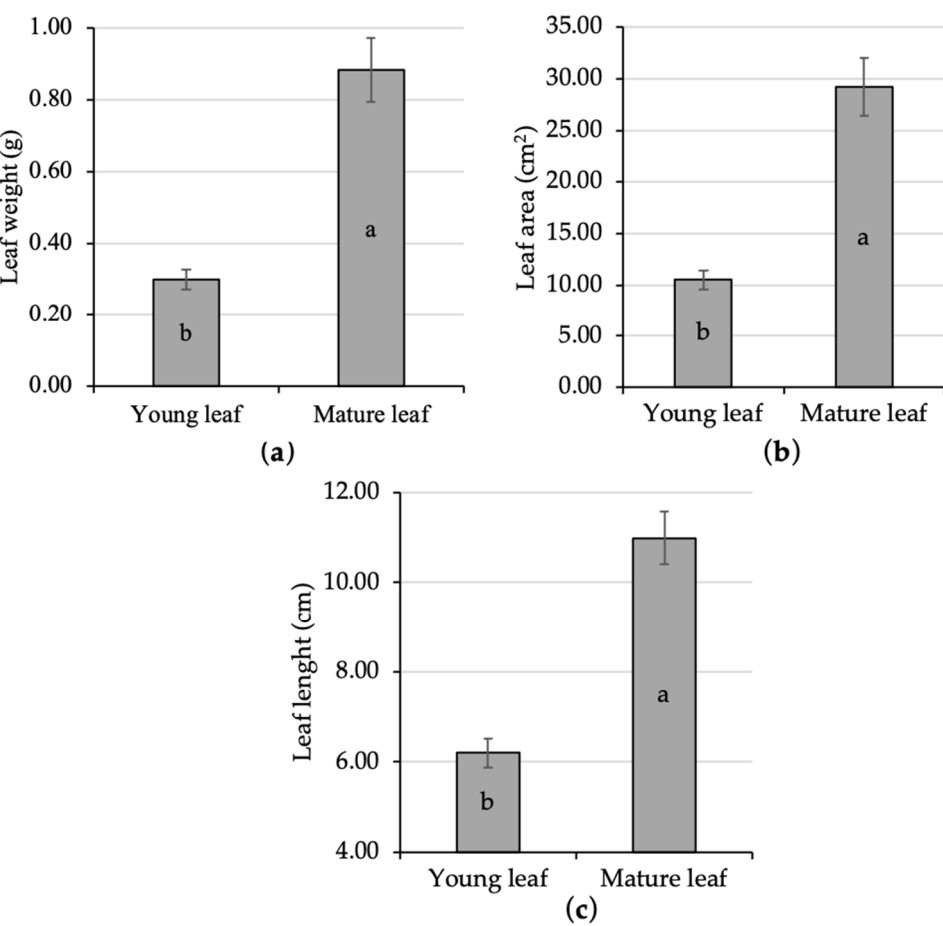

**Figure 1.** Comparative analysis of the size of young and mature kaffir lime leaf; namely, leaf weight (**a**), leaf area (**b**), and leaf length (**c**). Different letters inside the rectangular bar indicate a significant difference based on the LSD test at $\alpha$ 5%; the error bar represents the standard deviation.

### 3.2. Comparative Analysis of Leaf Eco-Physiological Characters

The portable photosynthesis system measured no significant difference in the eco-physiological characters regarding leaf temperature between young and mature leaves of kaffir lime. The variation in leaf temperature in the present study was 24.48–24.58 °C (Figure 2a). This range was still in the normal category, since an earlier study reported that 75% of net assimilation rate was carried out at 20–30 °C [48]. Above that regime was

categorized as high temperature, and the high temperature resulted in a restriction on citrus photosynthetic capacity [49].

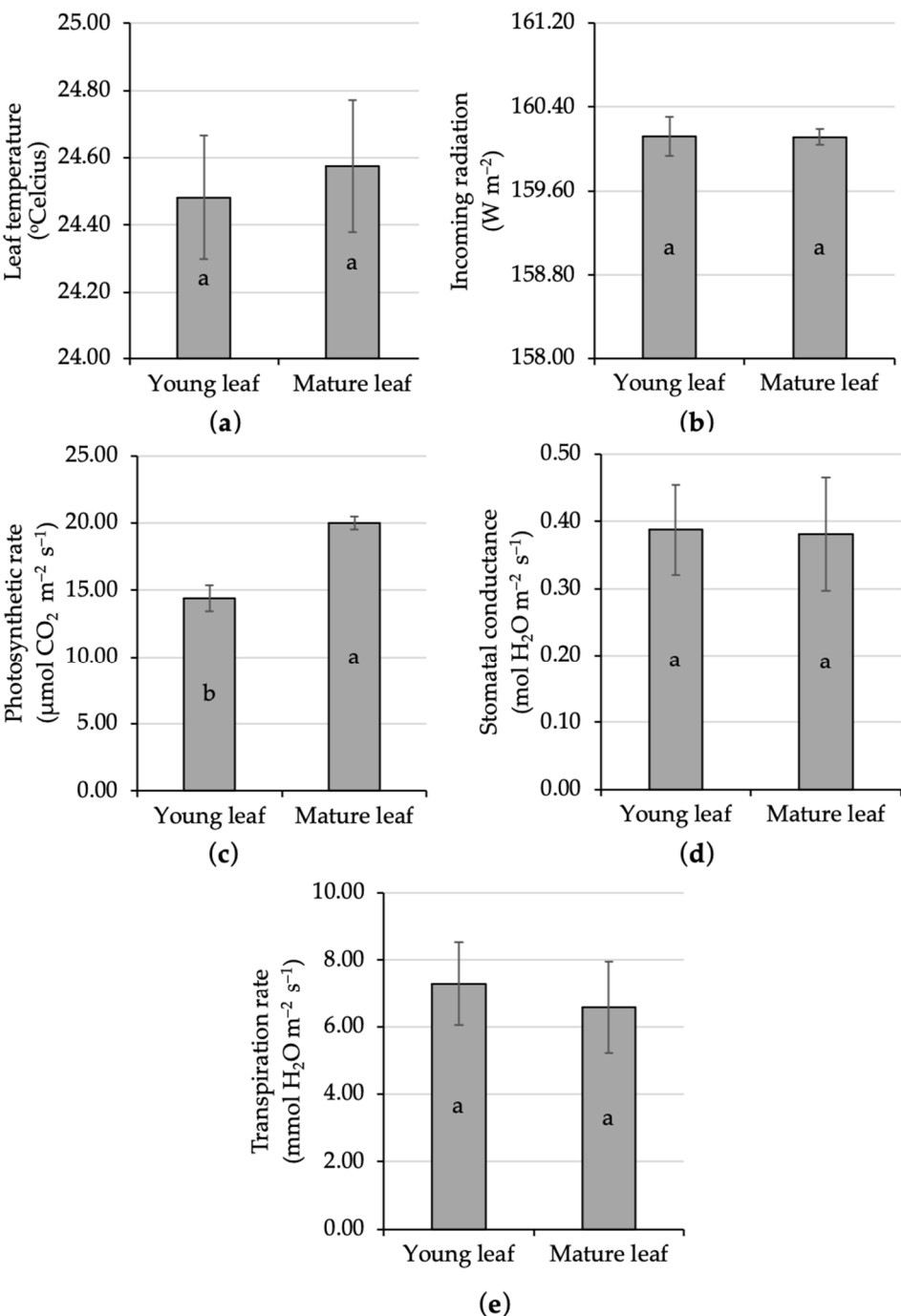

**Figure 2.** Comparative analysis of the ecophysiological characters of young and mature kaffir lime leaves; namely, leaf temperature (**a**), incoming radiation (**b**), photosynthetic rate (**c**), stomatal conductance (**d**), and transpiration rate (**e**). Different letters inside the rectangular bar indicate a significant difference based on the LSD test at α 5%; the error bar represents the standard deviation.

The leaf temperature was associated with the incoming radiation, since the sunlight not only becomes the main raw material for assimilation but also it could release some heat to the plant. Thus, incoming radiation was also interesting to reveal. The present finding found no significant difference in radiation exposed to young and mature leaves, i.e., 160 W m$^{-2}$ (Figure 2b). The same rate of incoming radiation here, emphasizes that if

there is a difference in the rate of photosynthesis between young and mature leaves, it is genuinely caused by the age of the leaves.

Leaf age significantly affected the photosynthetic rate of kaffir lime under full sun conditions. The photosynthetic rate on mature kaffir lime leaf had been significantly increased for about 39% compared to the young one (Figure 2c). This finding was in agreement with the common argument that photosynthetic rate lagged behind the expansion of a leaf [50]. Numerous studies reported the lower photosynthetic rate on immature unexpanded young leaves compared to mature ones, namely, lychee [51], mango [50], apple [52], avocado [53], and other fruit crops [54]. Fully developed mature leaves with a dark green color in the exposed canopy was responsible to be the main important source organ that supported the whole plant growth and development [51]. The relationship between photosynthetic rate and leaf age did not always form a linear pattern, because there was proof that showed a logarithmic reduction in the photosynthetic rate up to 50% in older leaves [55]. Aside from leaf age, the citrus photosynthetic rate was also influenced by the nutritional status, i.e., boron (B)-adequate leaves, B-deficient leaves, and B-toxicity leaves display different photosynthetic rates of 15 $\mu mol\ m^{-2}\ s^{-1}$, 2 $\mu mol\ m^{-2}\ s^{-1}$, and 6 $\mu mol\ m^{-2}\ s^{-1}$, respectively [56]. In addition, the variation in foliage photosynthetic rate might also relate to stomatal diffusion conductance of $CO_2$ [48].

Statistical analysis showed no significant difference in stomatal conductance between young and mature kaffir lime leaves, i.e., 0.38–0.39 mol $H_2O\ m^{-2}\ s^{-1}$ (Figure 2d). Stomatal conductance, the opposite of stomatal resistance, was an important variable that displayed the degree of opening of the stomata allowed gas exchange and water release/transpiration. No significant stomatal conductance was followed by a similar result in terms of the transpiration rate. Both young and mature kaffir lime leaves displayed a relatively similar transpiration rate, in the range of 6.58–7.29 mmol $H_2O\ m^{-2}\ s^{-1}$ (Figure 2e). The higher the stomatal conductance, the more the stomata opened, and the greater potential to have high transpiration and photosynthetic rates [57]. Stomatal conductance could be varied in response to genetic and environmental factors. Different citrus species might show different stomata sizes and densities; for example, *C. hystrix* had bigger stomata and a lower stomata density than *C. limon* and *C. aurantifolia* [58]. Environmental stress, such as drought, caused the increase in stomatal closure. Earlier studies have reported that stomatal closure is the first action made to deal with drought stress [59], resulting in a decline in net assimilation [60–62].

The present study also highlighted the effect of leaf age on water-use efficiency at the individual leaf level. The actual WUE of the young leaf experiences a significant increase up to 53% when it was getting mature, i.e., 2.08 up to 3.18 $\mu mol\ CO_2$ mmol $H_2O^{-1}$ (Figure 3a). This finding is in accordance with a previous study [2] that stated that WUE was positively related to tree age. The variation in WUE is commonly found in response to different species, time measurements, and culture practices. An earlier study [63] reported that the lower leaf water-use efficiency of Carrizo citrange (*Citrus sinensis* (L.) Osbeck), compared to Cleopatra mandarin (*Citrus resnhi* Hort. ex Tanaka; Cleo), was caused by different genetic factors. On a daily basis, the WUE was clearly high in the morning, gradually decreasing up to the afternoon [60]. Culture practices such shading [64] and irrigation modification [65–67] were reported to gain plant WUE. A reduction in over-optimal irrigation to optimal irrigation, forming a slight water deficit [68], was reported to improve the WUE and citrus yield by 30% and 20%, respectively [2].

It was likely that the mature leaves were also more efficient to use light than the young leaves, i.e., 12.68 and 4.3 $\mu g\ lux^{-1}$, respectively (Figure 3b). There was a triple gain in LUE in mature leaves compared to young ones. The optimal LUE and photosynthetic rate could vary from species to species and also occurred in a specific range of temperature [49]. Although both young and mature leaves were exposed by similar incoming radiation and expressed the same leaf temperature, there is the supposition that young leaves have other limiting factors so that the LUE and photosynthetic rates were difficult to extend. This condition might be associated with the presence of non-stomatal limiting factors, i.e., the

limited capacity of chloroplast to absorb $CO_2$ [59]. Therefore, there is a need to analyze the leaf pigment content and its correlation to the leaf assimilation rate for providing some important details.

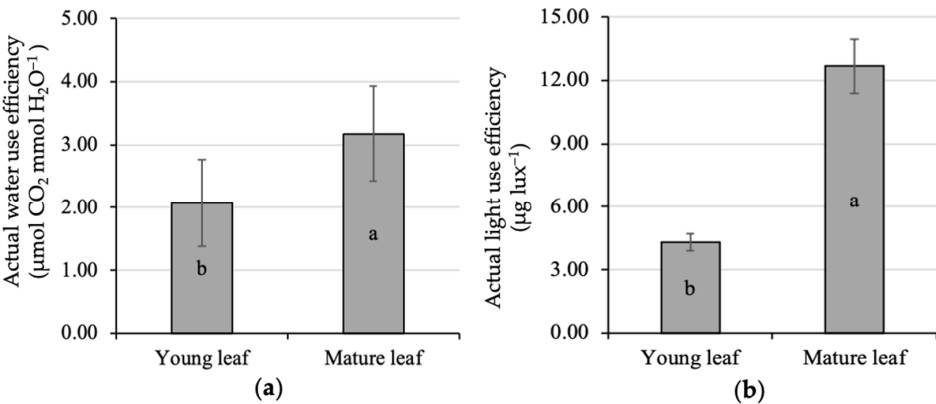

**Figure 3.** Comparative analysis of the actual water (**a**) and light-use efficiency (**b**) of young and mature kaffir lime leaves. Different letters inside the rectangular bar indicate a significant difference based on the LSD test at α 5%; the error bar represents the standard deviation.

### 3.3. Comparative Analysis and Correlation of Leaf Pigmentation

Leaf pigment in the present experiment was statistically different given the effect of leaf maturation. The mature leaf experienced significant improvement in all measured pigments, i.e., chlorophyll-a, chlorophyll-b, chlorophyll-total, carotenoid, and anthocyanin, of about 321%, 285%, 311%, 171%, and 29% respectively, compared to the younger ones (Figure 4). A similar finding was also reported in the coffee leaf maturation process [45,69]. However, both chlorophyll-a and chlorophyll-b were heat and high-temperature sensitive compared to carotenoid [70]. Carotenoid is also a plant pigment essential for fruit coloring [71] and light harvesting during environmental stress [72]. The presence of a high carotenoid content could help the process of heat dissipation of excess excitation energy in the photosystem [73]. Aside from the photo-protection effect, this pigment serves as an antioxidant, phytohormone precursor, and color attractant agent [74]. The relationship of leaf pigment and photosynthetic rate is interesting to further evaluate.

Correlation analysis is a statistical tool to find the strength and direction of relationships among several variables data. Numerous studies have used this tool to reveal the relationship among morphological, ecophysiological, and phytochemical variables of plant [7,8,14,75,76]. The present study used Pearson correlation analysis to reveal the strong, significant, and positive correlation between the photosynthetic rate and all leaf pigment variables, except anthocyanin [Table 1]. The higher leaf chlorophyll-a, chlorophyll-b, chlorophyll total, and carotenoid contents were followed by the higher leaf photosynthetic rate and vice versa. It was likely that the higher content of chlorophyll and carotenoid in mature leaves is the main reason behind the higher photosynthetic rate found in the same leaf. A younger leaf is known to have a lower chlorophyll content, so that less photosynthesis occurs. Additionally, young leaves are more sensitive to heat and excessive light stress due to the lower carotenoid content produced.

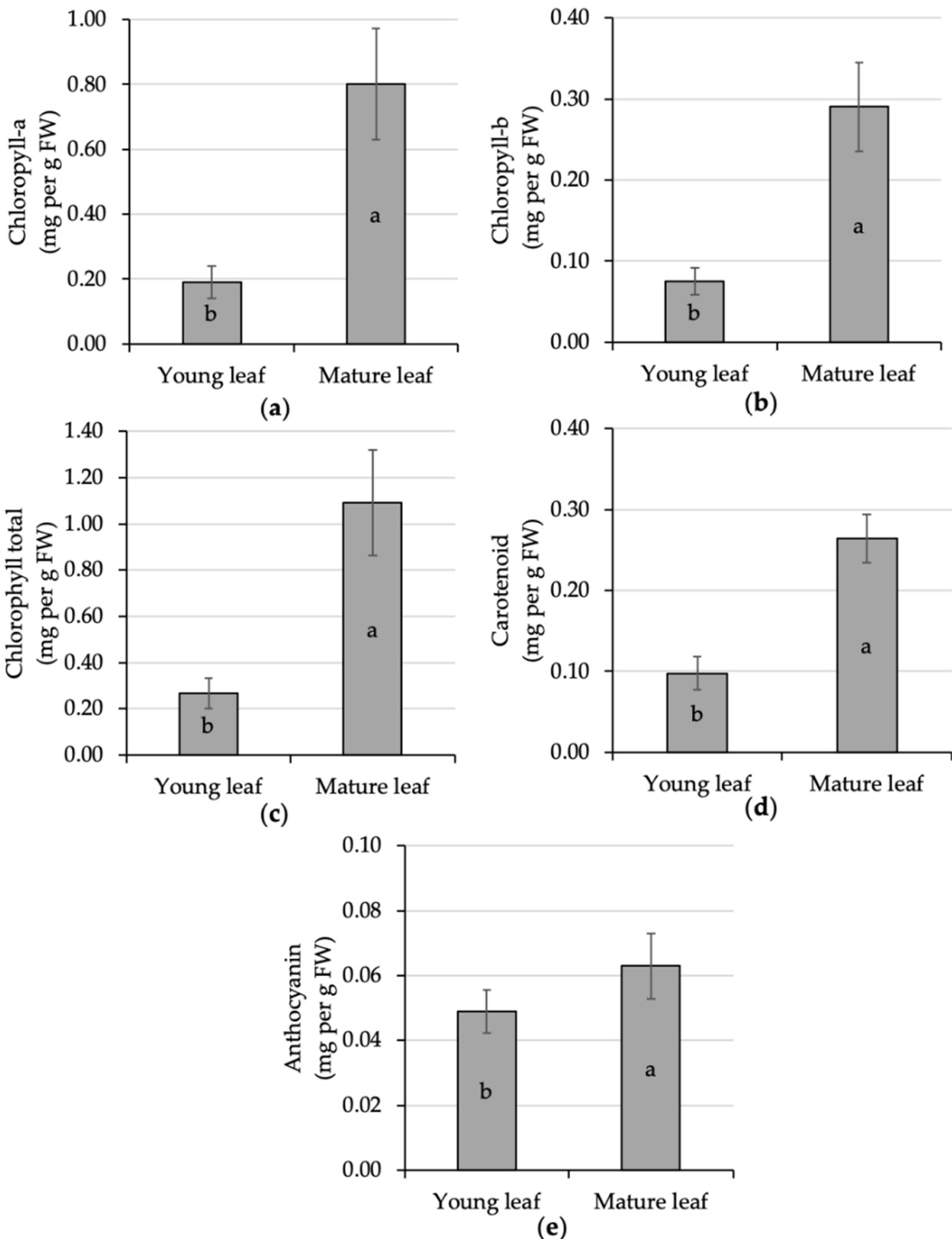

**Figure 4.** Comparative analysis of the pigment content of young and mature kaffir lime leaves; namely, chlorophyll-a (**a**), chlorophyll-b (**b**), chlorophyll total (**c**), carotenoid (**d**) and anthocyanin (**e**). Different letters inside the rectangular bar indicate a significant difference based on the LSD test at α 5%; the error bar represents the standard deviation. Leaf pigment is expressed per unit leaf fresh weight (FW).

**Table 1.** Pearson correlations among the leaf pigments and photosynthetic rate of kaffir lime.

| Variable | Chl-a | Chl-b | Ant | Car | Chl-t |
|----------|-------|-------|-----|-----|-------|
| Chl-b | 0.9992 * | | | | |
| Ant | 0.5496 | 0.5744 | | | |
| Car | 0.9927 * | 0.9939 * | 0.6343 | | |
| Chl-t | 0.9999 * | 0.9996 * | 0.5561 | 0.9932 * | |
| Photo | 0.8777 * | 0.883 * | 0.6569 | 0.8946 * | 0.8792 * |

Note: Chl-a—Chlorophyll-a (mg per g FW), Chl-b—Chlorophyll-b (mg per g FW), Ant—Anthocyanin (mg per g FW), Car—Carotenoid (mg per g FW), Chl-t—Chlorophyll total (mg per g FW), Photo—Photosynthetic rate ($\mu$mol $CO_2$ m$^{-2}$ s$^{-1}$). *—significantly correlated at 99% confidence level.

## 4. Conclusions

Comparative analysis revealed the different leaf characters between young and mature leaves on one-year-old kaffir lime trees in open-field conditions. Young leaves experienced a significant improvement in leaf size and leaf photosynthetic rate, actual water-use efficiency, and light-use efficiency, as the leaf matured. Leaf photosynthetic rate displayed a positive and strong correlation with leaf pigment, except anthocyanin. In the future, the uses of the present finding include, but are not limited to, (i) formulating a regression model on leaf physiology and pigmentation; (ii) creating a culture practice recommendation to gain more mature leaves; and (iii) to better understand the science behind leaf maturation, when it could be combined with other studies in genomics, transcriptomics, proteomics, and metabolomics.

**Author Contributions:** Conceptualization, R.B. and R.P.; methodology, R.B. and E.S.; resources, R.P., A.A. and D.E.; writing—original draft preparation, R.B.; supervision, R.P., D.E., E.S. and A.A.; APC funding acquisition, R.B. All authors have read and agreed to the published version of the manuscript.

**Funding:** The APC was fully funded by Universitas Padjadjaran, Indonesia.

**Institutional Review Board Statement:** Not applicable.

**Informed Consent Statement:** Not applicable.

**Data Availability Statement:** Not applicable.

**Acknowledgments:** The authors thank Nandang Hasanudin for his technical assistance during Li-6400XT measurement and also the Pasir Kuda management team of IPB for field technical assistance.

**Conflicts of Interest:** The authors declare no conflict of interest.

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
