# Peer review of "Comparative and Correlation Analysis of Young and Mature Kaffir Lime (Citrus hystrix DC) Leaf Characteristics"

_2037-0164, doi:10.3390/ijpb13030023_

Round 1

Reviewer 1 Report

I would like to appreciate the authors for their effort in making this study; I think this is an important and interesting topic and may provide a better understanding on science behind the kaffir lime leaf maturation.

The abstract abides by all the editing instructions and presents very clear the objectives of the study. The introduction is well written and supported by well selected bibliographic data. All bibliographic sources are fairly recent and correctly mentioned in the text.

However, the following are recommended for manuscript improvement:

Line 59. I think it is important to specify the type of soil on which the kaffir lime trees were grown.

Line 115. Replace ″different alphabet″ with ″different letter″. Please check all Figures and modify as recommended.

Figure 4 is not mentioned in text. Please do this (probably at sub-item 3.3, Line 193).

Line 241. I recommend that you improve the Conclusions section, mentioning a concise conclusion that emerges from all these results. Because, for now, what you present in the Conclusions section is actually a resumption of the results obtained.

Lines 293, 304, 351, 380, etc. Please write in italics the scientific name. Please check the entire manuscript and modify as recommended.

Reviewer 2 Report

Dear authors,

I had a great opportunity to review your research manuscript entitled "Comparative Analysis and Correlation of Young and Mature Kaffir Lime (Citrus hystrix DC) Leaf Characteristics".

Below I list several questions and comments about the manuscript that I believe will improve it.

*Graphical  Abstract : if it's possible, please add a graphical absatract to describe rapidely you experimental plan 

*Title

Please improve the title of the article it's not understandble 

*Materials methodes,

 Research Procedure : Please make more clear your experimental plan! 

Resultas and discussion

For the mesurement of chlorophyll content please precise in the figure 4,

g per FW , Idem for table 1 

3.3. Future Potential Studies: put this part in conculsion without subtitle 

Good luck,
